# Anomalous criticality coexists with giant cluster in the uniform forest model

Hao Chen[1*], Jesús Salas[2,3†] and Youjin Deng[4,5‡]

**1** School of the Gifted Young, University of Science and Technology of China,
Hefei, Anhui 230026, P.R. China
**2** Departamento de Matemáticas, Universidad Carlos III de Madrid,
Avenida de la Universidad 30, 28911 Leganés, Spain
**3** Grupo de Teorías de Campos y Física Estadística, Instituto Gregorio Millán Barbany,
Universidad Carlos III de Madrid, Unidad Asociada al Instituto de Estructura de la Materia,
CSIC, Spain
**4** Department of Modern Physics, University of Science and Technology of China,
Hefei, Anhui 230026, P.R. China
**5** Hefei National Laboratory,University of Science and Technology of China,
Hefei, Anhui 230088, P.R. China
* chenhao123@mail.ustc.edu.cn, † jsalas@math.uc3m.es, ‡ yjdeng@ustc.edu.cn

November 10, 2023

## Abstract

We show by extensive simulations that the whole supercritical phase of the three-dimensional uniform forest model simultaneously exhibits an infinite tree and a rich variety of critical phenomena. Besides typical scalings like algebraically decaying correlation, power-law distribution of cluster sizes, and divergent correlation length, a number of anomalous behaviors emerge. The fractal dimensions for off-giant trees take different values when being measured by linear system size or gyration radius. The giant-tree size displays two-length scaling fluctuations, instead of following the central-limit theorem.

# 1 Introduction

Percolation studies connectivity in random geometric systems [1,2]. In statistical physics, percolation has been of immense theoretical interest, providing a simple example that undergoes a non-trivial phase transition. The celebrated Ising and Potts models [3] can be described as a correlated percolation through the exact Fortuin-Kasteleyn transformation [4–6]. Thanks to the percolation approach, it was established [7–11] that the Ising model in three dimensions (3D) has a sharp continuous phase transition, and in 4D, it exhibits mean-field critical behavior, proving the triviality of the 4D Euclidean scalar quantum field theory. Percolation has also intensively been studied in mathematics, including a list of variations like $k$-core and explosive percolation, etc [12–17]. Also, percolation has been applied in diverse branches of science and industry [18, 19].

In the basic bond percolation, one randomly occupies each lattice edge with probability $p$ and constructs clusters of connected components. Clusters are small for small $p$, and the probability (two-point correlation) that two sites with distance $r$ are in the same cluster decays exponentially as $g(r) \sim \exp(-r/\xi)$, and the correlation length $\xi$ diverges as $\xi \sim (p_c - p)^{-\nu}$ as threshold $p_c$ is approached. At $p_c$, the size $s$ of fractal clusters follows a universal power-law distribution as $n(s) \sim s^{-\tau}$, and the correlation decays algebraically as $g(r) \sim r^{2-d-\eta}$. For the supercritical phase ($p > p_c$), an infinite cluster of size $C_1$ occupies a nonzero fraction of the lattice—i.e., $m \equiv C_1 L^{-d}$ converges to a constant as the linear size $L \to \infty$. In percolation, $m$ plays a role as the order parameter and behaves as $m \sim (p - p_c)^{\beta}$ as $p \downarrow p_c$. Further, the second moment of cluster sizes, $\chi \equiv L^{-d} \sum_i |C_i|^2$, acting as the magnetic susceptibility, diverges as $\chi \sim |p - p_c|^{-\gamma}$. Among these critical exponents $\nu, \beta, \gamma, \eta, \tau$, two are independent and the others can be obtained from (hyper)scaling relationships [20].

For $p > p_c$, the infinite cluster can be even proved to be unique—i.e., there is one and only one giant cluster. All off-giant clusters are small with finite correlation length $\xi'$, and the correlation $g'(r)$, with the giant cluster being excluded, vanishes exponentially. In 2D, the supercritical and subcritical ($p < p_c$) phases are dual to each other. Further, the smallness of $\xi'$ predicts that the size fluctuation of the giant cluster would obey the central-limit theorem and follow a normal (Gaussian) distribution.

In this work, we study the percolative properties of the supercritical phase for the (weighted) uniform forest (UF) model in 3D [21–24]. The UF model (also called the arboreal gas) consists of a spanning forest of trees (acyclic clusters), in which each tree is weighted by a factor $w$ per occupied bond; the statistical weight of any configuration $\mathcal{A}$ can be written as $\pi(\mathcal{A}) = w^{|\mathcal{A}|} \delta_{c(\mathcal{A}),0}$. This is similar to that for bond percolation with probability $p = w/(1+w)$, except the $\delta$-function constraint on zero cyclomatic number $c(\mathcal{A}) = 0$. Further, as bond percolation, the 3D UF model undergoes a continuous transition at $w_c$ [Fig. 1(a)]. It was found that $w_c = 0.43365(2)$, $\nu = 1.28(4)$, and $\beta/\nu = 0.4160(6)$ [25]. Moreover, the supercritical phase also has an infinite and unique tree [26], similar to percolation. As shown in the inset of Fig. 1(a) for $w = 0.9 > w_c$, $m$ quickly saturates to $m_0 = 0.685(6)$, and is clearly long-ranged.

# 2 Results

Despite these analogues to bond percolation, we find that the whole supercritical phase of the 3D UF model exhibits the simultaneous emergence of a surprisingly rich variety of critical behaviors and of a unique giant tree, providing a counter example for the standard percolation scenario.

In the supercritical phase, the off-giant clusters are fractal and display critical scaling behaviors that are generally expected from percolation theory at criticality. (i) As shown in

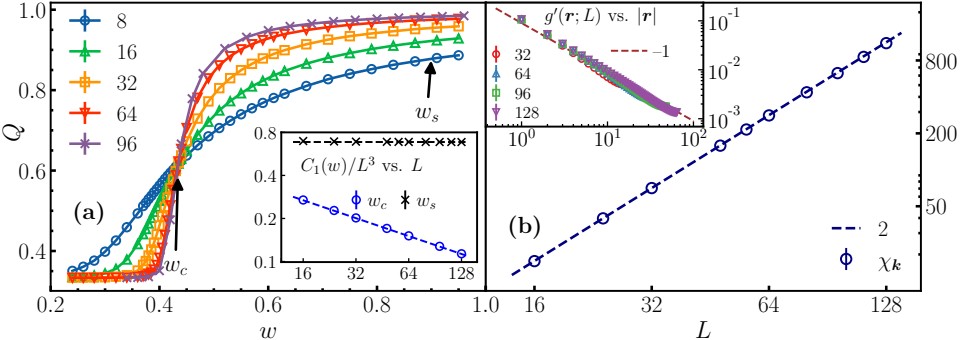

Figure 1: Simultaneous emergence of critical behaviors and a giant cluster. (a) The percolation transition and the giant tree of size $C_1$ in the supercritical phase. The approximately common intersection of the ratios $Q$ for different sizes $L$ indicate the threshold at $w_c \approx 0.43365$. The inset shows that, while the order parameter $m(w_c) \equiv C_1 L^{-3}$ algebraically vanishes, it quickly saturates to a constant, which is $m_0 = 0.685(6)$ for $w_s = 0.9$. (b) Emergent critical behaviors at $w = w_s$ demonstrated by the power-law divergence of the Fourier-transformed susceptibility $\chi_k \sim L^{1.99(2)} = L^2$, and the algebraic decaying of the off-giant correlation $g'(r) \sim |r|^{2-d}$.

Fig. 1(b), the Fourier-transformed susceptibility $\chi_k$ (the definition will be given later) diverges as $\chi_k \sim L^{\gamma/\nu} = L^{1.99(2)}$. This result is in good agreement with $\chi_k \sim L^2$, following the standard finite-size scaling (FSS) *Ansatz*. The off-giant correlation $g'(r)$ algebraically decays as $g'(r) \sim |r|^{2-d-\eta}$ with $\eta = 0$ [24]. Indeed, the scaling relation $2 - \eta = \gamma/\nu$ is satisfied. (ii) The size distribution of the clusters $n(s, L)$ has two terms: one accounts for the contribution of the off-giant trees, while the other takes care of the giant one. The former term contains a power law $s^{-\tau}$ times the size distribution of the off-giant clusters $\tilde{n}'(s, L)$, which is governed by the second-largest cluster of size $C_2 \sim L^{d_{C_2}}$ with $d_{C_2} = 2.29(2)$. The latter term has a prefactor $L^{-d}$, and it is governed by the size of the largest cluster $C_1 \sim L^{d_{C_1}} = L^{3.000(2)} = L^d$. Putting all together, we have [see Fig. 2(a)]:

$$n(s, L) = s^{-\tau} \tilde{n}'(s L^{-d_{C_2}}) + L^{-d} n_1(s, L). \tag{1}$$

The Fisher exponent is $\tau \equiv \tau_2 = 2.31(2)$, and the distribution peak, arising from $C_1$, defines an effective Fisher exponent $\tau_1 = 2$. The hyperscaling relations $\tau_i = 1 + d/d_{C_i}$ for $i = 1, 2$ are satisfied.

In addition, the supercritical phase exhibits a variety of unusual critical behaviors. (iii) The standard FSS theory predicts that the critical correlation length is $\xi \sim O(L)$, and, indeed, the gyration radius $R_1$ of the largest tree scales as $R_1 \sim L$ for $w \geq w_c$. However, for $w > w_c$, the off-giant correlation length, characterized by the gyration radius $R_2$ of the second-largest cluster, diverges sublinearly as $R_2 \sim L^{\kappa_2}$ with $\kappa_2 = 0.76(2)$ [see the inset of Fig. 2(b)]. This indicates that the supercritical phase has two length scales—i.e., $L$ and $L^{0.76} \ll L$. Typically, the size $s$ of a fractal object depends on its gyration radius $R$ as $s \sim R^{d_f}$, and the generic fractal dimension $d_f$ of the giant cluster coincides with the finite-size fractal dimension $d_{C_1}$ since $R_1 \sim L$. (iv) For the off-giant clusters at $w > w_c$, however, the generic and finite-size fractal dimensions take different values—e.g., $d_{C_2} \neq d_{f_2}$ for the second-largest cluster. Instead, one has $d_{f_2} = d_{C_2}/\kappa_2$, since $C_2 \sim L^{d_{C_2}} \sim R_2^{d_{f_2}}$ and $R_2 \sim L^{\kappa_2}$. We obtain $d_{f_2} = 3.01(8) \approx 3$, which agrees well with $d$ and $d_{C_1}$; this is further demonstrated for all the off-giant clusters in Fig. 2(b). Thus, the off-giant clusters share the same generic fractal structure as the giant one.

The giant cluster, occupying about 70% of the lattice for $w = 0.9$, exhibits interesting critical behaviors. In a traditional supercritical phase with finite correlation length, the central

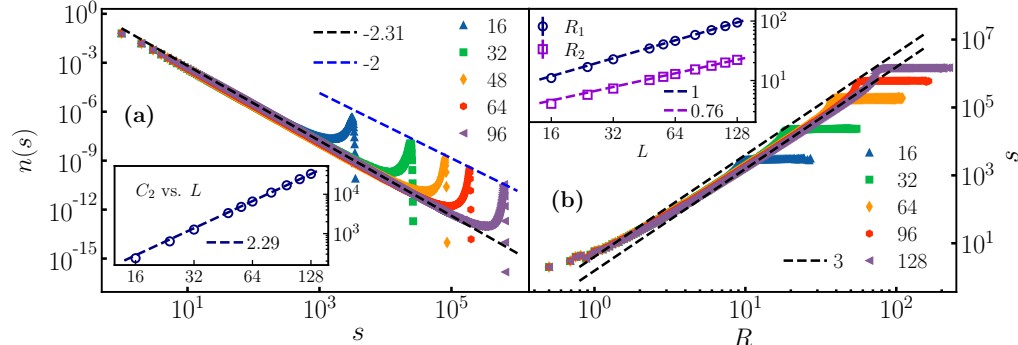

Figure 2: Fractal structures and two-length scales at $w = w_s$. (a) Scaling behavior of the cluster-size distribution (1). The inset displays $C_2 \sim L^{d_{C_2}}$ and with a line of slope $d_{C_2} = 2.29$. (b) Power-law dependence of cluster size $s$ on the gyration radius $R$. The curves seem to collapse around two parallel lines of slope $d_f = 3$. The inset shows the radii of the two largest clusters, of which the fits give exponents $\kappa_1 = 0.999(4)$, and $\kappa_2 = 0.76(2)$.

limit predicts that the giant-cluster size should follow a normal (Gaussian) distribution and the normalized fluctuation $F_1 \equiv \mathrm{Var}(C_1) L^{-d}$ should converge to some constant. (v) As in Fig. 3(b), however, we find $F_1 \sim L^{d_{F_1}}$ with $d_{F_1} = 2.03(6)$. This is counter intuitive because one would naively expect the central-limit theorem to apply because the ratio $\xi'/L \sim L^{\kappa_2 - 1} \to 0$ as $L \to \infty$ (here $\xi'$ is the off-giant correlation length).

We then consider the probability density function (PDF) $f_{C_1}(C_1, L)$ of the random size $C_1$ of the giant cluster in a lattice of linear size $L$. In the standard FSS theory, by rescaling $\mathcal{X} = (C_1 - \langle C_1 \rangle) L^{-d_{C_1}}$, and transforming $f_{\mathcal{X}}(x) dx = f_{C_1}(C_1, L) dC_1$, one can obtain a universal and $L$-independent function $f_{\mathcal{X}}(x)$. (vi) In the supercritical phase of the UF model, however, we find that the $f_{C_1}(C_1, L)$ data for different values of $L$ cannot be collapsed onto a unique curve by any single exponent like $d_{C_1} = 3$. Thus, we consider the probability $f_{\mathcal{X}_1}(x_1, L) dx_1$ for the rescaled random deviation $\mathcal{X}_1 \equiv (C_1 - \langle C_1 \rangle) L^{-d_{C_2}}$ with $d_{C_2} = 2.29$. Then, the $f_{\mathcal{X}_1}(x_1, L)$ data approximately collapses well near $x_1 = 0$ and for $x_1 > 0$ [Fig. 3(a)]. Nevertheless, $f_{\mathcal{X}_1}(x_1, L)$ has a wide-range shoulder for $x_1 \ll 0$, for which an approximate data collapse can be achieved by $L^{\delta} f_{\mathcal{X}_1'}(x_1') dx_1' \equiv f_{\mathcal{X}_1}(x_1, L) dx_1$ with $\mathcal{X}_1' \equiv (C_1 - \langle C_1 \rangle) L^{-3}$ and $\delta = 0.77$. This means that the whole configuration space is roughly partitioned into two sectors: one takes up a finite configuration-space volume while the other vanishes asymptotically as $L^{-\delta}$. In the dominant sector, the critical fluctuation of $C_1$ is governed by $d_{C_2}$ for off-giant clusters. In the vanishing sector, the variance $\mathrm{Var}(C_1)$ is $\sim O(L^{2d})$. Note that this exponent takes the largest possible value. We further sample $\mathrm{Var}(C_1)$ conditioned on $C_1 - C_1 \geq 0$ for the dominant (dom) sector, and $L^{-d}(C_1 - C_1) \leq a$ for the vanishing (van) sector, where we take $a = -0.1$. We obtain $d_{F_1}(\text{van}) = 2.96(4)$ and $d_{F_1}(\text{dom}) = 1.58(2)$ [see Fig. 3(b)], the latter of which gives $d_{C_2} = 2.29(1)$ from relation $2d_{C_2} - 3 = d_{F_1}$. Note that $d_{F_1}(\text{total}) = 2.03(6)$ for the total configuration space is distinct from $d_{F_1}(\text{dom})$ or $d_{F_1}(\text{van})$, indicating that the crossover regime also plays an important role.

## 3   Theoretical Insights

The critical behaviors of the 3D UF model can be partially understood from its relation to statistical mechanical systems and from the perspective of quantum field theory. The UF model corresponds to the $q \to 0$ limit of the $q$-state Potts model [3, 27–29] in the Fourtuin–

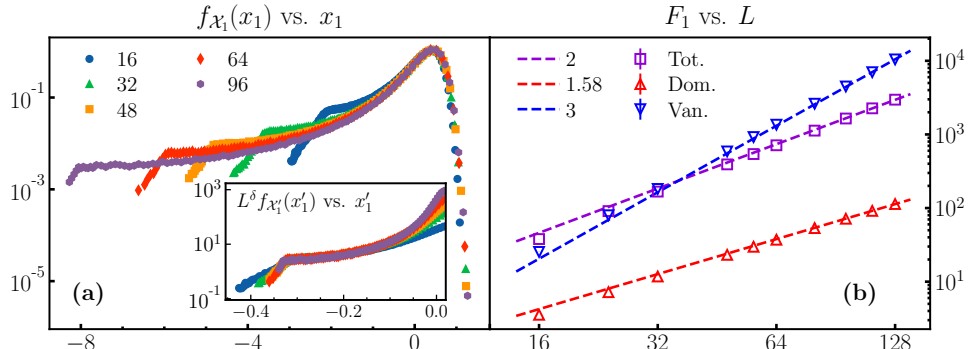

Figure 3: Two-length-scale fluctuation of $\mathcal{C}_1$ at $w = w_s$. (a) PDF $f_{\mathcal{X}_1}(x_1, L)$ of the rescaled random variable $\mathcal{X}_1 \equiv (\mathcal{C}_1 - \langle \mathcal{C}_1 \rangle) L^{-d_{\mathcal{C}_2}}$ with $d_{\mathcal{C}_2} = 2.29$. A good collapse of data is observed in the peak region. The inset shows the PDF $L^{\delta} f_{\mathcal{X}_1'}(x_1')$ of the random variable $\mathcal{X}_1' \equiv (\mathcal{C}_1 - \langle \mathcal{C}_1 \rangle) L^{-3}$ with $\delta = 0.77$, showing an approximate collapse in the shoulder region. (b) Normalized variance $F_1 = \mathrm{Var}(\mathcal{C}_1) L^{-3}$ in different sectors. The fits give exponents $d_{F_1}(\text{total}) = 2.03(6)$, $d_{F_1}(\text{dominant}) = 1.58(2)$, and $d_{F_1}(\text{vanishing}) = 2.96(4)$.

Kasteleyn random-cluster representation [4–6]. Particularly, this limit should be taken such that $v/q \equiv (e^J - 1)/q = w$ is held fixed ($J$ is the reduced nearest-neighbor coupling).

Caracciolo *et al.* [30] mapped the UF model onto a non-Gaussian fermionic theory with the non-abelian continuous OSP(1|2) supersymmetry by generalizing Kirchhoff's matrix-tree theorem. They also showed how to map this model in perturbation theory to all orders in $1/w$ onto a $N$-vector model analytically continued to $N = -1$. They concluded that in 2D, the model is asymptotically free, implying the absence of phase transition for any finite $w > 0$ [22, 23]; the criticality occurs at $w = +\infty$ [28, 31]. The relation of the UF model and the supersphere non-linear sigma model with $\mathbb{S}^{0,2}$ was studied in [32–34].

Recently, the UF model was interpreted [23] as a non-linear sigma model with the fermionic hyperbolic plane $\mathbb{H}^{0|2}$ as the target space. The hyperbolic symmetry of spin models has many interesting properties [35], and for the UF models, the relation leads [24] to the existence of a percolation transition at finite $w_c > 0$ for $d \geq 3$. It was also proven [26] that the supercritical phase has, in the infinite-lattice limit, a unique infinite tree for $d = 3, 4$. Moreover, it was established [24] that, at large enough $w > 0$, there are massless power-law correlations for $d \geq 3$, as if the model were at criticality,

$$g(\boldsymbol{r}) = g_0 + c\,|\boldsymbol{r}|^{2-d} + \cdots, \tag{2}$$

where $g_0 = m^2$ comes from the giant cluster, $c = c_0 + O(1/w)$ is a constant, and the dots stand for higher-order corrections. Nevertheless, despite Eq. (2), the percolative properties in the supercritical phase remain elusive.

## 4 Algorithms and observables

We use the Sweeny algorithm [36] and simulate the UF model on the simple-cubic lattice with periodic boundary conditions, for $8 \leq L \leq 128$. At every step, one randomly picks up an edge $e_{ij}$ between sites $i$ and $j$. If $e_{ij}$ is occupied, the bond is removed with probability $\min(1, 1/w)$. If $e_{ij}$ is empty, it is occupied with probability $\min(1, w)$ if $i$ and $j$ belong to different trees, and, otherwise, it is left unoccupied since a bond on $e_{ij}$ would generate a cycle. The nontrivial operation is to detect the connectivity of $i$ and $j$ in a dynamical setting. Using the link-cut

tree data structure [37], the connectivity query can be efficiently implemented in $O(\log L)$ amortized time. Note that no critical slowing down occurs in 3D [25, 36, 38].

For a random configuration, we denote the forest of trees as $\{\mathcal{C}_k\}$, and specifically leave $\mathcal{C}_1$ and $\mathcal{C}_2$ for the largest and second-largest clusters, respectively. For a tree $\mathcal{C}_k$, an arbitrary site is chosen as the origin, and the "unwrapped" coordinate $\boldsymbol{x}_k^i$ of each site $i$ is obtained by growing the tree from the origin. This coordinate $\boldsymbol{x}_k^i$ is well defined, since the path connecting any two sites in a tree is unique. The mass-center coordinate $\overline{\boldsymbol{x}}_k = (1/\mathcal{C}_k) \sum_{i \in \mathcal{C}_k} \boldsymbol{x}_k^i$, and the squared gyration radius $\mathcal{R}_k^2 = (1/\mathcal{C}_k) \sum_{i \in \mathcal{C}_k} (\boldsymbol{x}_k^i - \overline{\boldsymbol{x}}_k)^2$ are calculated.

By detecting the connectivity between sites $i$ and $j$ over configurations, we measure the two-point correlation $g(\boldsymbol{r} = \boldsymbol{r}_i - \boldsymbol{r}_j)$, as well as the off-giant correlation $g'(\boldsymbol{r})$, where $\boldsymbol{r}_i$ is the standard Euclidean coordinate of site $i$. For simplicity, we choose $\boldsymbol{r} = (r, 0, 0)$ along the $x$-axis. Moreover, to study the algebraic decaying behavior of the correlation, an auxiliary Ising spin $s_i \in \{\pm 1\}$ is introduced for every site $i$: Independently for each tree, we assign the same value $s_i = 1$ or $-1$ with equal probability to all the sites in the tree. By definition, $g(\boldsymbol{r}_i - \boldsymbol{r}_j) = \langle s_i s_j \rangle$. The magnetization $\mathcal{M} = \sum_m s_m$ and its Fourier transform $\mathcal{M}(\boldsymbol{k}) = \sum_m s_m \exp(i \boldsymbol{k} \cdot \boldsymbol{r}_m)$ are sampled, where the summation is over the whole lattice. The smallest nonzero momenta in the $x$ direction, $\boldsymbol{k} = (2\pi/L, 0, 0)$ is used for simplicity.

Statistical average and probability distribution are then taken over the configurations generated in simulations—e.g., $C_k \equiv \langle \mathcal{C}_k \rangle$ and $R_k \equiv \langle \mathcal{R}_k \rangle$ for $k = 1, 2$. Also, we define the normalized fluctuation $F_1 \equiv \mathrm{Var}(\mathcal{C}_1) L^{-3}$, the susceptibility $\chi \equiv \langle \mathcal{M}^2 \rangle L^{-3}$, the dimensionless ratio $Q \equiv \langle \mathcal{M}^2 \rangle^2 / \langle \mathcal{M}^4 \rangle$, and the Fourier-transformed susceptibity $\chi_{\boldsymbol{k}} \equiv \langle \mathcal{M}(\boldsymbol{k}) \mathcal{M}(-\boldsymbol{k}) \rangle L^{-3}$.

## 5 Fits

As a powerful quantity for locating a continuous phase transition, the crossings of the $Q(w)$ curves for diffent sizes $L$ [Fig. 1(a)] clearly support the previously determined percolation threshold $w_c = 0.43365(2)$ [25]. We carry out extensive simulation at $w_c$ and $w_s = 0.9$, deeply in the supercritical phase. The critical behaviors, shown in Figs. 1–3, have been qualitatively presented in *Results*.

We perform the least-squares fits to a power-law *Ansatz* for any observable $\mathcal{O}(L)$

$$\mathcal{O}(L) = L^{d_\mathcal{O}} \left( a_0 + a_1 L^{-\omega_1} + a_2 L^{-\omega_2} \right) + b_0. \tag{3}$$

In most cases, we set $\omega_1 = 1$ and $\omega_2 = 2$, and $b_0 = 0$ is fixed for observables that vanish for $L \to \infty$. As a precaution against FSS corrections not included in the *Ansatz* (3), we have performed each fit by allowing only data with $L \geq L_{\min}$. By studying how the estimates of the parameters, as well as the $\chi^2$ per degree of freedom, vary as a function of $L_{\min}$, we determine our final estimates and their error bars. In particular, we consider the sizes and radii of the largest and second-largest clusters, the normalized fluctuation $F_1$, and the Fourier-transformed

|       | $d_{C_1}$  | $\kappa_1$ | $d_{C_2}$ | $\kappa_2$ | $d_{F_1}$ | $d_{\chi_k}$ |
|-------|-----------|-----------|-----------|-----------|-----------|-------------|
| tot.  | 3.000(2)  | 0.999(4)  | 2.29(2)   | 0.76(2)   | 2.03(6)   | 1.99(2)     |
| dom.  | 3.002(3)  | 1.001(5)  | 2.28(2)   | 0.78(2)   | 1.58(2)   | 1.63(3)     |
| van.  | 2.997(4)  | 0.997(6)  | 3.00(2)   | 1.01(2)   | 2.96(4)   | 2.83(5)     |

Table 1: Estimated critical exponents for the supercritical phase at $w_s = 0.9$, for the total (tot.) configuration space, and for the dominant (dom.) and vanishing (van.) sectors.

susceptibility $\chi_k$, which scale as ($j = 1, 2$)

$$C_j \sim L^{d_{C_j}}, \ R_j \sim L^{\kappa_j}, \ F_1 \sim L^{d_{F_1}}, \ \chi_k \sim L^{d_{\chi_k}}. \tag{4}$$

At the critical value $w_c$, the scaling behaviors follow the standard FSS theory, which predicts $\kappa_1 = \kappa_2 = 1$, $d_{C_1} = d_{C_2} = d_f$ ($d_f$ is the generic fractal dimension), and $d_{F_1} = 2d_f - d$. There is only one non-trivial exponent $d_{C_1}$, which is determined to be $d_{C_1} = 2.5840(6)$.

In the supercritical phase with $w = w_s$, the final estimates are given in Table 1. As expected, the largest cluster, occupying a finite fraction of the lattice, has trivial exponents $d_{C_1} = 3.000(2) = 3$ and $\kappa_1 = 0.999(4) = 1$. The effective Fisher exponent $\tau_1$, governing the decreasing of the distribution peak in Fig. 2(a), is also trivial $\tau_1 = 1 + d/d_{C_1} = 2.000(1)$.

The finite-size fractal exponent of the second-largest cluster is $d_{C_2} = 2.29(2)$, which has not yet been reported to our knowledge. This gives the Fisher exponent $\tau_2 = 1 + d/d_{C_2} = 2.31(2)$ in Eq. (1). The gyration radius scales sublinearly versus $L$ with exponent $\kappa_2 = 0.76(2)$, unexpected from the standard FSS theory. The generic fractal dimension, $C_2 \sim R_2^{d_{f_2}}$, is calculated as $d_{f_2} = d_{C_2}/\kappa_2 = 3.01(8) = 3$. Surprisingly, $d_{f_2}$ is just the spatial dimension, and Fig. 2(b) further gives $d_{f_2} = 3$ for all the off-giant clusters.

By definition, the Fourier-transformed susceptibility is $\chi(\boldsymbol{k}) = L^{-3} \sum_{m,n} \langle s_m s_n \rangle e^{i\boldsymbol{k} \cdot (\boldsymbol{r}_m - \boldsymbol{r}_n)}$, where the contribution from the background term $g_0$ in Eq. (2) is eliminated. Thus, $\chi(\boldsymbol{k}) \sim L^2$ is expected, and this is strongly supported by the estimated exponent $d_{\chi_k} = 1.99(2)$. Interestingly, the normalized fluctuation of the largest tree is governed by exponent $d_{F_1} = 2.03(6) = 2$.

Despite the simplicity of scalings like $C_1 \sim L^3$ and $F_1 \sim L^2$, the distribution of $\mathcal{C}_1$ is sophisticated [Fig. 3(a)]. Thus, we perform separate least-squares fits for the dominant and the vanishing sectors, respectively conditioned on $\mathcal{C}_1 - C_1 \geq 0$ and $\mathcal{C}_1 - C_1 \leq -0.1 L^3$. The results in Table 1 suggest that the two sectors have dramatically different scaling behaviors. Particularly, in the vanishing sector, we find that both the largest and the second-largest clusters have $d_{C_j} = 3$ and $\kappa_j = 1$ for $j = 1, 2$. Further, $d_{\chi_k} = 2.83(5)$ suggests that the $r$-dependent decaying of correlation $g(r)|_{\text{van}}$ is extremely slow (i.e., $g(r)|_{\text{van}} \sim r^{-0.17}$). This behavior, together with $d_{C_j} = 3$, gives a strong hint for a logarithmic decay: $g(r)|_{\text{van}} \sim 1/\log(r)$.

# 6 Conclusion

While undergoing a typical continuous percolation transition, the 3D UF model exhibits a variety of critical behaviors in the supercritical phase. The simultaneous existence of anomalous criticality and of a unique giant cluster is unexpected from the standard percolation theory. The critical scaling behaviors not only appear in the off-giant clusters, but also in the fluctuation of the giant tree. Unlike a conventional critical point, the whole configuration space can be approximately divided into two configuration sectors of distinct critical exponents. Further, the overall scaling behaviors arise from some delicate interplay of the two sectors and of the crossover regime in between.

Some insight can be borrowed from the fermionic field theory, but a complete and deep understanding is still needed for this extremely rich critical behavior. As a return, we believe that our work may also bring some insight for critical phenomena in the Potts model and the nonlinear-sigma model, which are two important classes of systems in statistical mechanics and condensed-matter physics. For instance, for the XY model with long-range interaction, which was recently found [39, 40] to exhibit critical behaviors in the low-temperature phase, it would be desired to study such critical behaviors from percolation perspective.

Some open quesions arise. For instance, what is the upper spatial dimensionality $d_u$ for the supercritical-phase criticality for the UF model? It is known that the zero-temperature

UF model (the uniform tree model) has $d_u = d_c = 4$ and, at criticality, it has $d_u = d_p = 6$. Thus, $d_c = 4$ or $d_p = 6$ can equally serve as a candidate of $d_u$ for the supercritical UF model. From recent studies for the Fortuin-Kasteleyn representation of the Ising model [41, 42], we may have the third scenario that the low-temperature UF model has simultaneously two upper dimensions at both $d_c = 4$ and $d_p = 6$.

## Acknowledgements

This work was initiated by private communications with Tyler Helmuth, Roland Bauerschmidt, and Nicholas Crawford, to whom we are indebted.

**Funding information** H.C and Y.D. have been supported by the National Natural Science Foundation of China (under Grant No. 12275263), the Innovation Program for Quantum Science and Technology (under grant No. 2021ZD0301900), Natural Science Foundation of Fujian province of China (under Grant No. 2023J02032). J.S. was partially supported by Grant No. PID2020-116567GB-C22 AEI/10.13039/501100011033, and by the Madrid Government (Comunidad de Madrid-Spain) under the Multiannual Agreement with UC3M in the line of Excellence of University Professors (EPUC3M23), and in the context of the V PRICIT (Regional Programme of Research and Technological Innovation).

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
