# Peer review of "Anomalous criticality coexists with giant cluster in the uniform forest model"

_SciPost Physics_

## Round 1 · Referee Report · Anonymous (Referee 1) · 2024-2-27

Strengths

1- interesting scaling behavior 2-connections with other models

Weaknesses

1- rationale for theoretical understanding of the results

Report

In this manuscript, the authors analyze the percolation (scaling) properties of a supercritical 3D weighted uniform forest model (UF). This model has similarities with other models in statistical physics, including Bernoulli bond percolation when the probability to occupy a lattice edge assumes a specific value (conditioned to be acyclic), or the q->0 limit of the q-state random cluster model. Since the model is connected to several supersymmetric spin models, it has been the object of recent interest in the physics community. By using the Sweeny algorithm on a simple-cubic lattice, the authors found several interesting features in the supercritical phase, including the emergence of two distinct sets of critical exponents and a scaling behavior that depends on the crossover between the corresponding configuration sectors. The authors tried to understand the critical behavior of their model, but what is reported in the manuscript are just references to other works using fermionic theory and their conclusions. While their results are interesting and certainly deserve publication, according to my opinion they should make some additional effort to improve the section of the theoretical insights gained from them.

Requested changes

1- improve the section of theoretical insights

  • validity: good
  • significance: good
  • originality: ok
  • clarity: good
  • formatting: good
  • grammar: good

Author:  Hao Chen  on 2024-04-03  [id 4386]

(in reply to Report 1 on 2024-02-27)

Dear Prof. Boninsegni and referee,

Thanks a lot for reviewing the manuscript and for the helpful feedback!
We have changed the manuscript according to the point raised in the report.
Please find below the detailed reply to the report.

Submitting respectfully,
The authors

[Response to Referee]

The referee writes:
"The authors tried to understand the critical behavior of their model, but what is reported in the manuscript are just references to other works using fermionic theory and their conclusions. While their results are interesting and certainly deserve publication, according to my opinion they should make some additional effort to improve the section of the theoretical insights gained from them."

Reply:
Thanks for pointing out that the "Theoretical Insights" section could be improved. This section aims to provide the readers with a background of the uniform forest (UF) model, including the mappings between the UF and various non-linear sigma models, and the corresponding theoretical predictions obtained from the latter field theories. As mentioned in the text, a central result from this mapping is the power-law correlation in the supercritical phase, which hints that there is also criticality in the supercritical phase. However, this connection is unable to provide more precise descriptions of the critical geometric properties of the supercritical phase, such as the scaling behaviors of the off-giant clusters, the cluster-size distribution, and the probability distribution of the giant cluster. Moreover, it is not clear whether the standard finite-size-scaling theory would apply to the criticality of the supercritical phase, as it is based on the assumption that the correlation length is O(L) for a finite system of side length L. Our simulations provide insights into these unknown questions, which are detailed in the "Results" section.

Therefore, we have expanded this section by carefully distinguishing the existing conclusions from the unknown questions mentioned above. We also moved the "Theoretical Insights" section before the "Results" section to make the paper more readable.

---

## Editorial Decision

resubmitted